# The “Hitchhiker’s Guide to the Galaxy” of Endothelial Dysfunction Markers in Human Fertility

**DOI:** 10.3390/ijms22052584

**Published:** 2021-03-04

**Authors:** Daniele Santi, Giorgia Spaggiari, Carla Greco, Clara Lazzaretti, Elia Paradiso, Livio Casarini, Francesco Potì, Giulia Brigante, Manuela Simoni

**Affiliations:** 1Department of Biomedical, Metabolic and Neural Sciences, University of Modena and Reggio Emilia, 42121 Modena, Italy; carlagreco@unimore.it (C.G.); clara.lazzaretti@unimore.it (C.L.); elia.paradiso@unimore.it (E.P.); livio.casarini@unimore.it (L.C.); giulia.brigante@unimore.it (G.B.); manuela.simoni@unimore.it (M.S.); 2Unit of Endocrinology, Department of Medical Specialties, Azienda Ospedaliero-Universitaria of Modena, 41125 Modena, Italy; spaggiari.giorgia@aou.mo.it; 3International PhD School in Clinical and Experimental Medicine (CEM), University of Modena and Reggio Emilia, 42121 Modena, Italy; 4Center for Genomic Research, University of Modena and Reggio Emilia, 42121 Modena, Italy; 5Department of Medicine and Surgery-Unit of Neurosciences, University of Parma, 43121 Parma, Italy; francesco.poti@unipr.it

**Keywords:** endothelium, endothelial dysfunction, male fertility, female fertility, reproduction

## Abstract

Endothelial dysfunction is an early event in the pathogenesis of atherosclerosis and represents the first step in the pathogenesis of cardiovascular diseases. The evaluation of endothelial health is fundamental in clinical practice and several direct and indirect markers have been suggested so far to identify any alterations in endothelial homeostasis. Alongside the known endothelial role on vascular health, several pieces of evidence have demonstrated that proper endothelial functioning plays a key role in human fertility and reproduction. Therefore, this state-of-the-art review updates the endothelial health markers discriminating between those available for clinical practice or for research purposes and their application in human fertility. Moreover, new molecules potentially helpful to clarify the link between endothelial and reproductive health are evaluated herein.

## 1. Sex Differences in Endothelial Function Phenotype

Human fertility is controlled by hormones constituting the hypothalamic–pituitary–gonadal (HPG) axis, aiming to promote gametogenesis in both males and females [1]. The action of sex steroids is strictly connected to the endothelial cell functioning, which is crucial in regulating reproductive system homeostasis. The endothelium is a monolayer of cells forming the inner lining of all blood vessels with protective properties essential to maintain physiological vascular functions upon balancing vasoconstriction and vasodilation stimuli [2]. Dysfunctions of the single-cell layer may contribute to the pathogenesis of vascular disease [3,4] and could lead to detrimental effects for human fertility.

Since the occurrence of cardiovascular diseases (CVD) is higher in men than women [5,6,7], endothelial dysfunctions could impact fertility in a sex-specific manner and could be indicative of gender-related differences of the hormonal milieu. The risk of stroke, coronary artery and peripheral vascular diseases is very low in young women and increases with age, achieving maximal levels after the menopause [8], similar to those described in men. This gender-related diversity was imputed to estrogens, since both endogenous levels of estradiol and the expression of its receptors differ considerably between sexes [9,10,11,12]. In line with this view, during women’s fertile window, the cyclicity of estradiol levels is demonstrated as cardiovascular-protective, inducing total and low-density lipoprotein (LDL) cholesterol reduction [13], inhibiting LDL oxidation [14], reducing fibrosis, stimulating angiogenesis and vasodilation, and improving mitochondrial functions [15]. Estradiol directly regulates endothelial function and homeostasis inducing vasodilation by activating the transcription of endothelial nitric oxide (NO) synthase (eNOS) and upregulating NO production [15,16]. Indeed, physiological changes in estradiol serum levels during menstrual cycle are directly related to flow-mediated dilation (FMD) [17]. During menopause, the decline of estrogen levels corresponds to increased CVD risk in females, achieving similar levels in males [18]. Since vascular smooth muscle functionality does not change during menopausal age, these data suggest that increased CVD risk is linked to the loss of sex hormone regulation on endothelial cell functioning [19].

From a molecular point of view, the impact of estrogen on the endothelium could involve three different mechanisms: (i) induction of vasodilator factors such as NO and prostacyclin, (ii) stimulation of endothelial damage repair systems, and (iii) anti-inflammatory and antioxidant properties [20,21] (Figure 1).

Estrogenic actions are mediated via genomic (long-term) and non-genomic (short-term) pathways depending on the receptor involved [10,21]. Although the estrogen action on endothelial cell is largely studied, the androgens effects are less examined [9]. Testosterone exerts a vasodilator effect via both genomic and non-genomic actions on the endothelial cell [22]. The genomic effect is mediated by the nuclear androgen receptor (AR)-dependent increase in NO production, activating the eNOS [23,24] (Figure 1). The non-genomic effect requires the action of four membrane receptors [25,26], activating the phosphoinositide 3-kinase (PI3K)/protein kinase B (AKT) pathway and eNOS [23,24]. Although the exact molecular mechanisms induced by testosterone on endothelial cells is not completely understood [22], its vascular action is direct and not mediated by estrogens synthesized through testosterone aromatization, as demonstrated in vitro [27,28,29,30,31]. These data are strengthened by studies demonstrating the vasodilation effect of dihydrotestosterone (DHT), a non-aromatizable androgen [32,33]. The testosterone-induced vasodilation needs the integrity of the vascular endothelium, while higher concentrations of the hormone are required to induce a dilatory response of dysfunctional endothelium [34]. In summary, sex differences in endothelium functions may be influenced by the action of both estradiol and testosterone, that clearly show sex-specific serum concentrations. Other circulating molecules, such as microRNAs [35,36,37,38], contribute to gender-related endothelial functions since they are differently distributed between premenopausal women and age-matched men [39].

This evidence confirms the gender-related difference in endothelial functions/dysfunctions which has to be considered in the context of human fertility. In this review, we summarize evidence available on the interconnection between male and female fertility and endothelial function, with a specific focus on the direct and indirect markers available to measure it.

## 2. How to Measure Endothelial Dysfunction: Direct and Indirect Parameters of Endothelial Health

### 2.1. Dynamic Markers

Endothelial dysfunction, expressed as a worsened capacity of the vessels to respond to physical and chemical stimuli, could be evaluated using several direct and indirect parameters aiming at detecting the principal endothelial mediator (i.e., NO). These markers can be measured using both invasive and non-invasive approaches. The angiographic measure of vascular diameter change in response to intracoronary infusions of acetylcholine (Ach) provides the most accurate currently available test [40]. Ach induces the NO release from the endothelial cell, in turn inducing vasodilation. Since the angiographic evaluation is an invasive practice, the non-invasive “Ach test” was developed. This protocol compares the response to an endothelium-independent vasodilator, such as nitroprusside, to that inducible by Ach [41]. Briefly, the venous occlusion plethysmography at forearm is used to compare the blood flow after Ach and sodium nitroprusside infusions [41]. However, since the clinical application of this technique is burdened by high costs and limited availability, further indirect measures have been developed, such as the FMD. This latter consists of the high-frequency ultrasonography measure of the brachial artery dilation after an induced hypoxia [42,43]. Using an arm cuff inflated to supra-systolic blood pressure level, the NO release from endothelial cells mediates relaxation of smooth muscle cells with subsequent vasodilation [42,43]. The FMD is expressed as the difference percentage between arterial diameter measured at baseline and after cuff releasing [42,43]. Since FMD is correlated with the NO concentration [44], measurable during coronary angiography via invasive procedures, FMD assessment remains the most used non-invasive parameter in clinical practice and research [45]. Two recent meta-analyses confirmed the FMD clinical relevance, predicting peripheral arterial disease [46], CVD [47,48], and all-cause mortality [47,49,50,51,52].

Alongside vascular elasticity, arterial stiffness is largely demonstrated to be related with CVD and mortality. Thus, the assessment of arterial stiffness in several proximal and distal vascular compartments has been suggested so far [53]. To this purpose, the most reliable parameter is the intima–media thickness (IMT), measured applying the linear probe longitudinally on arteries, searching for a double-line density between the intimal-luminal and the medial-adventitial interfaces [54]. Carotid IMT (cIMT) has been demonstrated to be associated with atherosclerosis, CVD [55,56], and erectile dysfunction. The latter is a precocious marker of endothelial dysfunction [57,58]. Indeed, subjects with erectile dysfunction present higher cIMT and lower FMD than controls, and these parameters may be improved upon vasodilation induced by treatment with phosphodiesterase 5 inhibitors [59,60,61,62]. Beyond cIMT, the aortic IMT (aIMT) is considered a potential marker of early atherosclerosis in children, adolescents, and young adults [63]. Interestingly, the aIMT has been found associated to impaired Sertoli cell markers, such as anti-Müllerian hormone and inhibin B, in obese adolescent men with insulin resistance [64]. Together with cIMT and aIMT, the arterial stiffness could also be measured in brachial (bIMT) and common arteries (caIMT) (Table 1) [65].

Since the endothelial dysfunction is characterized by reduced endothelial NO production, resulting in increased peripheral arterial resistance [66], pulse wave velocity (PWV) is a further vessels stiffness marker. Carotid-femoral PWV (cfPWV) indicates the velocity of propagation of the arterial blood pressure wave along the vascular wall [67] and represents the gold standard for arterial stiffness assessment [68]. This parameter has been demonstrated to predict organ damage [69] and CVD mortality [70], although it is still poorly applied in clinical practice.

Even magnetic resonance imaging (MRI) has been increasingly used for assessing endothelial functioning [71]. Quantitative, dynamic, non-contrast MRI studies have been developed, including the blood oxygen level-dependent (BOLD) imaging, the arterial spin labelling (ASL), the phase contrast (PC), the MRI-measured PWV, the luminal FMD, and the dynamic MRI oximetry [71]. These innovative approaches provided the opportunity to measure endothelial dysfunctions in human sites not easily accessible. For instance, MRI may be used for evaluating the NO-mediated retinal vessels vasodilation, as an indirect marker of the vascular elasticity in the microcirculation [72]. The dynamic MRI oximetry is applied to assess the flicker light-induced retinal vasodilatation after pupil dilatation induced by 1% topical tropicamide. The vasodilation is indicated as the percentage change in retinal vessel diameter from the baseline [73]. Moreover, the arterial elasticity could be measured in the kidney by the duplex ultrasound-derived parameter “renal resistive index” (RI) [74] and the dynamic renal RI [75]. The latter provides the RI measure after sublingual nitrate administration and is correlated to endothelial function and arterial stiffness [75].

### 2.2. Biochemical Markers

Impairment of endothelial function correlates with all cardiovascular risk factors, in particular, dyslipidemia. Oxidized low-density lipoprotein (ox-LDL) would stimulate the endothelial secretion of adhesion molecules and chemoattractant proteins, enhancing the atherosclerosis plaque development [76,77]. Similarly, LDL and triglycerides have been associated to endothelial dysfunction in both males and females [78]. Accordingly, cholesterol-lowering therapies have been demonstrated to improve endothelial vasomotor function of brachial arteries analyzed by FMD [79,80,81]. On the other hand, high-density lipoprotein (HDL) is known to exert endothelial protective roles through the apolipoprotein A-I receptor [82], mediating the cholesterol and phospholipid efflux and reducing the atherogenic stimulus [83,84]. Moreover, HDL stimulates vasodilation via NO release, which underlies the activation of eNOS by the scavenger receptor B type I [85], and the transport of bioactive lysophospholipids regulating vascular tone via sphingosine-1-phosphate 3 receptor [86,87,88,89]. Thus, the measurement of these various components of lipid homeostasis could serve as an indirect marker of endothelial dysfunction. 

C-reactive protein (CRP), a hepatic acute-phase protein, is considered a prognostic factor for major CVD [90,91]. Moreover, increased CRP serum levels have been associated with the Ach test impairment [92]. Indeed, CRP decreases eNOS expression and bioactivity in human aortic endothelial and smooth muscle cells [93,94]. Accordingly, the CRP reduction correlates with improvement of the endothelial function [92]. In addition, high-sensitivity CRP (hs-CRP), a sensitive and more precise biomarker of inflammation, provides results associated with a higher risk of atherosclerosis and all-cause and CVD mortalities [95].

Homocysteine is a sulfur-containing, non-proteinogenic amino acid biosynthesized during the metabolism of the essential amino acid methionine [96]. Hyperhomocysteinemia is an independent risk factor for atherosclerosis, stroke, CVD [96,97], and endothelial dysfunction [98]. Accordingly, several studies have reported the correlation between total homocysteine concentration and FMD [99], cIMT [100], and cfPWV [101]. Indeed, high homocysteine plasma levels presumably affect NO bioavailability and impair endothelial-dependent vasodilation [102,103]. Thus, the panel of biochemical indirect markers of endothelial dysfunction available in clinical practice should combine the lipid asset with the measurements of hs-CRP and homocysteine (Table 1).

A recently proposed marker of endothelial dysfunction is the lipoprotein-associated phospholipase A2 (Lp-PLA2), an inflammatory mediator produced by macrophages in atherosclerotic plaques. Physiologically, about 80% of Lp-PLA2 circulates bound to LDL, whereas the remaining 20% to HDL. Lp-PLA2 levels are elevated in patients with endothelial dysfunction [104], suggesting they might have a pro-atherogenic role [105]. Lp-PLA2 hydrolyses ox-LDL to LDL, generating pro-inflammatory and pro-atherogenic molecules promoting vascular inflammation and atherosclerotic plaque development [105]. These data were corroborated by a clinical study enrolling a group of 172 subjects with no significant coronary artery disease [106]. In this study, high concentrations of Lp-PLA2 were demonstrated as a strong predictor of endothelial dysfunction assessed by the Ach test. However, discordant results have been obtained using FMD [107] and Ach test after the administration of a Lp-PLA2 inhibitor [108]. Thus, these contrasting results, together with the difficult to measure Lp-PLA2, limit the reliability and use of this parameter in clinical practice.

Several other biochemical markers of endothelial dysfunction are currently used exclusively for research purposes, but not in clinical practice. These parameters could be divided according to their pro- or anti-inflammatory properties, as well as vasoconstriction or vasodilation actions, and location in specific tissues, or solubilized in the plasma (Table 2).

Inflammation markers may be informative of the activation state of the vascular endothelium, which, in turn, influences the adhesion and infiltration of leukocytes and the initiation of the coagulation cascade [109]. Moreover, other molecules are used to evaluate the response to oxidative stress and of the endothelium recovery in response to harmful stimuli [109,110]. Vasoconstriction molecules are mainly involved in the renin–angiotensin–aldosterone system, regulating blood pressure, plasma volume, and arterial muscle tone [111,112]. On the contrary, vasodilation molecules are generally related to the molecular paracrine action of NO, bradykinin, and potassium ion channels [113].

## 3. Endothelial Function and Male Fertility: Endothelial Gene-Related Polymorphisms

The male reproductive system requires the interaction between neurons, blood vessels, and cells of the immune system, which act synergistically with the main actors, such as germ cells, stromal testicular cells, and hormones [114]. In this refined mechanism, a potential endothelium role is suggested. Indeed, a bidirectional relationship occurs between the endothelial function and those hormones involved in reproduction. From one side, the endothelium’s role in balancing vasodilation and vasoconstriction is fundamental for reproductive functions. On the other hand, the endothelium is demonstrated to directly secrete or indirectly mediate the production of several molecules able to influence testicular functions [114]. However, the complex relationship between endothelial function and the male reproductive system is still not completely clarified and only few endothelium-related health markers described above have been evaluated in the context of male fertility/infertility. A step forward may be provided by the study of single-nucleotide polymorphisms (SNPs) of genes encoding key factors for endothelial physiology. They might elucidate the potential role of these genes in the male infertility pathogenesis and address some cases of infertility currently defined as “idiopathic”. Table 3 and Figure 2 describe the role and the effect of these potential markers of male infertility.

### 3.1. ENOS Gene Polymorphism and Male Infertility

NO is a molecular mediator relevant to many physiological systems, including reproduction 114. Its production is regulated by NOS [115], available in three different isoforms: eNOS, neural NOS (nNOS), and inducible macrophage NOS (iNOS), depending on the originator cell population [116]. Among the three isoforms, eNOS is the most expressed in male reproductive organs, since it is synthesized in endothelial testicular cells, vas deferens, epididymis, and both Sertoli and Leydig cells [117]. In testicular tissues, eNOS regulates the NO local production, balancing the reactive oxygen species (ROS)/antioxidants ratio [118,119]. This fine mechanism is crucial for maintaining proper testicular functionality, since germ cells are very susceptible to elevated ROS levels [118]. NO is supposed to be inversely related to sperm motility, to attend sperm acrosome reaction and capacitation process, and to regulate the Sertoli/germ cells ratio in the seminiferous epithelium, allowing the production of mature and functional sperms [118,120,121,122]. It is widely demonstrated that SNPs falling within the *eNOS* gene, located on the chromosome 7q36, modulate the expression of the enzyme and, in turn, NO production [123]. Some of these SNPs, such as the c.T786C (rs2070744), falling in the promoter region, the c.G894T (rs1799983) on exon 7, and a variable number of intron 4 tandem “4a4b repeats” (rs61722009) 124, have been associated with fertility and sperm morphology.

In vitro and epidemiological studies evaluated the association between *eNOS* SNPs and infertility, obtaining inconclusive results [117,124,125,126,127,128,129,130]. It is largely demonstrated that the excess in NO production leads to a sperm number reduction, by inducing germ cells apoptosis 117, and a sperm motility impairment, via activation of the s-Guanylyl Cyclase (sGC)-cGMP-protein kinase G (PKG) pathway [118]. This molecular mechanism is supposed to impact RNA stability [131], although the increased NO production is directly linked to oxidative damage of lipids, proteins, and DNA of sperm cells [132]. The *eNOS* SNPs c.T786C and 4a4b would modulate the activation of this signalling pathway and were identified as risk factors for male infertility, due to their association with higher NO production [117]. However, another study found that a specific c.T786C and c.G894T allelic variant was significantly associated with impaired sperm number, motility, morphology, and seminal glutathione peroxidase [128]. A recent meta-analysis detected only c.T786C *eNOS* SNP as a predictor of semen alterations, i.e., sperm concentration and motility, in a population of 3507 men [123,132]. Despite the existence of pathophysiological rationale and experimental evidence, rigorous association studies and functional demonstrations of the association between eNOS SNPs and infertility are still lacking. Therefore, these missing points prevent the clear comprehension of the *eNOS* variants’ role in spermatogenesis and their clinical implications.

### 3.2. Antioxidant Systems Gene Polymorphisms and Male Infertility 

ROS are free radicals physiologically required in seminal plasma to promote sperm capacitation, acrosome reaction, and sperm–oocyte fusion. These effects are strictly dose-dependent, since relatively high ROS concentrations in seminal plasma induce detrimental consequences, such as sperm DNA damage, lipid peroxidation, membrane fluidity reduction and sperm motility impairment [133,134]. Effects of seminal ROS may be neutralized by the homeostatic antioxidant systems, to which belong manganese superoxide dismutase (SOD2), catalase (CAT), glutathione peroxidase 1 (GXP1), and glutathione S transferase (GST) [135]. This complex enzymatic kit was found also in endothelial cells [136]. In particular, SOD2 catalyzes the detoxification of superoxide radicals in H2O and O2 in mitochondria [136,137]. Although the gene sequences encoding these enzymes are phylogenetically conserved, SNPs could modulate their expression levels [136] and, consequently, interfere with the ROS/antioxidants balance, influencing male fertility [136,137].

Several studies have found an association between *SOD2* gene SNPs and male infertility. The *SOD2* promoter SNP -262C > T “T” allele (rs4880) was recently demonstrated to be more frequent in infertile men than in the control group 135. Moreover, the incidence of the Val16Ala “Val” homozygous allele is higher in infertile than fertile men, suggesting that the “Ala” carriers have higher seminal H_2_O_2_ levels and lower sperm quality [138]. Similar associations were achieved by the evaluation of the *SOD2* SNPs p.Val16Ala [138] and p.Ile58Thr [139]. However, the clear link between these genotype variants and phenotypical translations is still absent. For instance, a different distribution of the *SOD2* p.Ile58Thr SNP in fertile vs. infertile men was not detected by a previous study [135]. Finally, the *SOD3 c.*G362A SNP (rs2536512) was recently evaluated in 111 infertile men compared to 104 fertile controls [140], not detecting an association between genotypes and infertility, although a significant reduction in SOD activity was found in samples from the infertile group [140]. Thus, these three SNPs are promising candidates for explaining the role of SOD2 and SOD3 enzymes in male fertility, although further studies are needed to clearly address this issue.

Considering other antioxidant systems involved in ROS balance, other SNPs were studied, such as *CAT* c.C262T (rs1001179), *GXP1* p.Pro200Leu (rs1050450), and two human GST isozymes presenting a null allele as a result of a deletion (*GSTT* and *GSTM*) [135]. However, the current literature is scant to establish the role of these SNPs in male infertility. The *CAT* gene is located on chromosome 11p3, encoding for an enzyme involved in the detoxification of H2O2 to H2O [141]. Only one study suggested that the allelic variant *CAT* C-262T CT identified a favorable fertility phenotype, while the CC genotype was associated with a doubled risk of male infertility [135]. Accordingly, higher H2O2 concentration and a reduced CAT activity were detected in the seminal plasma of infertile than fertile men [135]. However, this result needs to be further confirmed by independent clinical studies evaluating the contribution of other SNPs in the pathogenesis of male infertility.

Overall, these studies, although not entirely conclusive, suggest that the allelic variant of genes implicated in oxidative stress and expressed at the endothelial level could impact parameters of male fertility.

## 4. Endothelial Function and Male Fertility: Inflammation Markers

Soluble adhesion molecules and pro-inflammatory cytokines are markers of systemic inflammatory status [142,143] and are involved in several pathophysiological processes [142,143,144]. Indeed, a link between the immune system and fertility parameter was proposed decades ago [145] and was recently confirmed by experimental data [146]. Endothelial cells are primary targets for several pro-inflammatory cytokines and mediate the interaction with circulating immune cells, regulating tissue infiltration. Although these mechanisms may be activated at the systemic level, a focus on their implications in the male reproductive system is provided in the following sections.

### 4.1. Cytokines and Semen Quality

The presence of leucocytospermia reduces semen quality and increases the sperm DNA fragmentation index [147,148]. This relationship is not completely understood, since a beneficial effect of moderate leucocytospermia in obtaining pregnancy has been suggested [149,150]. In any case, while the presence of leucocytospermia is generally considered a poor sensitive factor [151], cytokine seminal levels are considered a more accurate parameter to discriminate inflammatory and non-inflammatory semen [147]. Interleukin (IL)-8 is a cytokine produced by several cells, including blood and endothelial cells [152]. IL-8 acts synergistically with IL-1 to chemo-attract leucocytes in the inflammation site and to promote phagocytosis [153]. IL-8 concentration within the seminal plasma has been widely investigated as a possible marker of spermatogenesis [152,154]. However, its application as a predictive factor of the fertility status in clinical practice is still limited, since seminal IL-8 appeared directly related to leucocytospermia, while being inversely related to ejaculatory volume only in patients with male accessory gland infections [152]. In addition, conflicting evidence is available about the link between seminal IL-8 and semen parameters [155,156,157,158,159], which likely resulted as a possible indicator of alterations in the post-testicular male genital tract, rather than a marker of spermatogenesis.

Increased levels of interferon (IFN)-γ and IL-7 were found in the seminal plasma of men with chronic prostatitis and chronic pelvic pain syndrome [154]. In these patients, both IFN-γ and IL-7 could be related to reduced semen quality [154]. Moreover, other pro-inflammatory cytokines, such as IL-6, IL-1β, and tumor necrosis factor-alpha (TNF-α), were largely investigated both in blood and in semen samples of infertile males [160], providing contradictory results. Indeed, several studies described the connection between seminal impairment and increased levels of these molecules [160,161,162,163,164,165,166,167], while others did not [160,163,168,169]. Taken together, these data suggest that the activation of the inflammatory/immune response documented by the increase of pro-inflammatory mediators in the seminal plasma are mainly the mirror of an inflammatory status of the genital male tract, rather than an expression of a testicular inflammatory state. The link between inflammation and the measured male genital product, i.e., seminal fluid, remains still far from being elucidated.

### 4.2. Adhesion Molecules and Sperm Maturation 

In the testicular seminiferous tubules, the physiological micro-environment required for sperm development is preserved through dynamic reworking of cell junctions, constituting the blood–testis barrier [170]. Intercellular junctions are mainly formed by cadherins/catenins cell adhesion protein complexes and integrins, which connect proteins of the extracellular matrix to the intracellular cytoskeleton [171]. The main role of this barrier is the control of those molecules that could reach the intraluminal site and, in turn, the spermatogenic cells, limiting interferences in sperm maturation [170,172]. Indeed, there is evidence supporting that several chemotherapy agents induce spermatogonia differentiation arrest and the blockade of spermatocyte meiosis. These events lead to azoospermia, damaging the blood–testis barrier and inhibiting cell adhesion molecules (CAMs) and cadherin expression [170]. CAMs are integral membrane proteins typically constituted by three domains, i.e., extracellular, transmembrane, and intra-cytoplasmic, involved in inducing proper modifications of cell adhesion in response to physiological functional needs [171]. In particular, the CAMs extracellular domain has a Ig-like region able to mediate homophilic or heterophilic dimerization with integrins and cadherins [173]. The crosstalk between these cell adhesion systems supports the junction renovation in human testis [171]. Several members of CAMs family acting within the testis have been identified and studied in knock-out animal models, providing specific fertility phenotypes [171]. Among these CAMs, ICAM-1, its soluble form (sICAM-1), and VCAM-1 are the most investigated molecules in the context of male fertility.

ICAM-1 is expressed in Sertoli cells and germ cells in a stage-specific manner modulating blood–testis barrier properties. In particular, the over-expression of the integral membrane form reinforces the barrier function, while the soluble form (sICAM-1) downregulates several blood–testis barrier protein components [171,174]. The balance between ICAM-1 and sICAM-1 opposite action impacts the permeability of the blood–testis barrier [171]. Moreover, ICAM-1 recruits VCAM-1 at the endothelial cell level, mediating the cell adhesion signaling [175]. Indeed, VCAM-1 facilitates the leukocyte adhesion to the vascular endothelium [176,177], while its active soluble form (sVCAM-1) mediates immune and inflammatory reactions at the peripheral level [178]. The complexity of this scenario is increased by the up-regulatory role exerted by several cytokines, such as TNF-α, IL-6 and IFN-γ, on ICAM-1 and VCAM-1 [179,180,181].

Another actor playing a potential role in the physiological sperm migration into seminiferous tubules seems to be ICAM-2 [171]. This molecule is constitutively expressed in epithelial and endothelial cells at the testicular level, where it mediates the conformational “stabilization” of actin molecules, inducing spermiation [180].

Vascular endothelial cadherin (VEC) is a single-pass transmembrane protein playing a crucial role in stabilizing the intracellular cytoskeleton and mediating cell–cell contacts [182]. Alongside its adhesive function, VEC is involved in maintaining the homeostasis of the seminiferous epithelial cycle, showing a stage-specific expression pattern at the testicular level [183]. More precisely, VEC may be involved in the acquisition of spermatid polarity within the seminiferous epithelium [182]. In male mice, the loss of VEC expression occurring together with the progression of sperm maturation stages leads to impairment of germ-Sertoli cell contacts, intra-lumen transition of immature spermatids and, subsequently, to low sperm count in the ejaculate [183].

Taken together, our knowledge of adhesion molecule functioning in testicular tissues was provided mainly by animal models. However, its role in human spermatogenesis needs further clarifications. On the other hand, it is undeniable that this issue is hardly surmountable, considering that in vivo studies investigating the role of these molecules in human tissues are hardly doable.

### 4.3. Role of Endothelin-1 and Sperm Function

In the male genital tract, endothelin-1 (ET-1) is detectable at testicular, epididymal and prostatic levels both in human and in rat males [184,185]. Within the testis, ET receptors were mainly found in Leydig cells, in which they promote testosterone secretion [186] In Sertoli cells, ET receptors are involved in micro-tubule contractility [187]. However, the most known ET-1 role was observed in the epididymis, where it induces smooth muscle contractions required for supporting the transit of immotile sperms through the excretory ducts [188]. This concept is reflected by the positive correlation between ET-1 levels and seminal fluid volume [189]. Although these data demonstrated that ET-1 is involved in the regulation of human fertility, no evidence is currently available suggesting a physiological direct relationship between ET-1 and parameters indicative of sperm functions [190]. A genetically modified mouse model with ET-1 overexpression had reduced testicular blood flow, but neither increased inflammatory status, nor impaired cell proliferation [184]. Hypothetically, ET-1 dysfunction could impair sperm progression through the genital tract, negatively impacting the viability of gametes in the ejaculate. To date, properly designed clinical studies are needed to clarify this mechanism in humans.

### 4.4. Vascular Endothelial Growth Factor-A and Testicular Function

Vascular endothelial growth factor-A-encoding (*VEGF-A*) gene transcription leads to multiple alternative splicing isoforms resulting in either anti- or pro-angiogenic properties GF [191,192]. In humans, *VEGF-A* is physiologically expressed in testis, seminal vesicles, prostate, measurable in the seminal fluid, and involved in the control of male reproductive functions [193]. VEGF-A stimulates angiogenetic processes and modulates the testicular blood vessels permeability via balancing of pro-angiogenic and anti-angiogenic isoforms [191,194]. In addition, this factor is proposed to be a regulator of the spermatogonia stem cell pool within the testis, balancing at the same time its self-renewal throughout premature differentiation and cell death [191]. Accordingly, loss of VEGF-A isoforms in mouse models determines the reduction of sperm number, due to the impairment of those mechanisms required for long-term maintenance of undifferentiated spermatogonia [195]. In vitro experiments suggested that the action of VEGF-A may be beneficial for some sperm motility parameters, in a concentration-dependent manner. Moreover, mouse models revealed that VEGF-A induces germ cells proliferation, promotes testicular regeneration by enhancement of the vascularization [196] and supports Leydig cell steroidogenesis [197]. Since both steroidogenesis and gametogenesis require an intact testicular microvascular compartment [193,197], it is not surprising that VEGF-A paracrine effects are required to maintain physiological testicular functions [197].

## 5. Endothelial Function and Female Fertility

While the link between endothelial dysfunction and female sex hormones is described in the literature as reported above, the impact of endothelial dysfunction on female fertility in humans is unclear. Mouse models suggested that estradiol exerts a number of vasoactive properties, beyond its physiological action on the reproductive axis [198]. Therefore, it could be hypothesized that estrogen may be involved in the regulation of endothelial functions supporting female reproduction.

### 5.1. Endothelial Dysfunction Markers in Premature Ovarian Insufficiency

Premature ovarian insufficiency (POI) is defined as ovarian function failure occurring before 40 years. Clinically, it is characterized by amenorrhea, sex steroid deficiency, and high gonadotropin serum levels [199]. The rapid decline in circulating estrogen levels exposes women to a number of complications, such as the increased risk for cardiovascular morbidity and mortality [200,201]. Endothelial functions could be related to the pathogenesis of POI, as suggested by the lower FMD, IMT, and endothelial progenitor cells (EPCs) in women affected by the disease compared to healthy controls [202,203]. The FMD improvement after oral, six months’ estrogen/progestogen cyclic treatment supports the hormone-dependent etiology of endothelial dysfunction in POI [202]. These results were confirmed by a clinical study demonstrating that POI women treated with estrogen replacement therapy had similar endothelial function compared to controls [204].

### 5.2. Endothelial Dysfunction Markers in Polycystic Ovary Syndrome

Polycystic ovary syndrome (PCOS) is the most frequent endocrine disorder in women worldwide [205]. The disease is clinically characterized by a combination of hyperandrogenism, anovulation, and metabolic disorders [206]. The latter could display both endothelial dysfunctions [207] and early signs of atherosclerosis, such as increased IMT and coronary calcium deposit [208]. Endothelial dysfunction may be related to tissue resistance to the insulin-induced vasodilation, as demonstrated in vivo [207]. Indeed, the role of insulin in modulating endothelial function is confirmed by the beneficial effect of metformin treatment. This drug increases the peripheral insulin sensitivity, improving the reactive hyperemia-peripheral arterial tonometry in subjects with abnormal endothelial functions [209]. The same effect is demonstrated in PCOS rat models, in which the increase of insulin sensitivity by treatment with metformin and the anti-androgen flutamide improves endothelial functions [210]. A systematic review confirmed the relationship between endothelial dysfunction and insulin resistance [211]. However, since not all women with both insulin resistance and PCOS experienced endothelial dysfunction, other pathogenic contributing mechanisms should be considered.

Alongside insulin resistance, the estrogens/androgens imbalance has been proposed as a co-causative factor of endothelial dysfunction in PCOS [207]. Serum androgen levels may be elevated in women with PCOS [212]. Androgens could lead to endothelial dysfunction in both lean and obese women with PCOS, as recently evaluated by the elevation of ET-B (an ET-1 receptor) in the same cohort of patients [213]. Moreover, PCOS women have reduced EPCs number and function, likely due to increased ROS levels and impaired insulin signaling [214]. Finally, other biomarkers of endothelial dysfunction are increased in women with PCOS, such as visfatin, VEGF, matrix metallopeptidase 9 (MMP9), high-mobility group box 1, pentraxin 3, and soluble lectin-like oxidized low-density lipoprotein receptor-1 [215].

### 5.3. Endothelial Dysfunction Markers in Endometriosis: The Role of Cytokines

Endometriosis is a frequent benign gynecological disorder characterized by the presence of endometrial tissue outside the uterine cavity, caused by pathogenic mechanisms not fully understood yet. Endometrial cell survival outside the uterus is potentially supported by the formation of new vessels. Thus, angiogenic growth factors and circulating EPCs could serve as biomarkers for the diagnosis and classification of endometriosis [216]. Indeed, women affected by endometriosis have higher peritoneal levels of several angiogenic cytokines, such as VEGF, IL-1, IL-6 and IL-8 [217]. This cytokine-rich peritoneal fluid may increase the mitotic activity of endothelial cells, indicating the role of these molecules in promoting angiogenesis, implantation, and growth of endometrial transplants [218].

Interestingly, women with endometriosis have high oxidative stress and systemic inflammation, together with a pro-atherogenic lipid profile, which could increase the risk of developing atherosclerosis. Accordingly, some studies demonstrated the presence of subclinical atherosclerosis in women affected by endometriosis, with regression after surgical removal of ectopic masses [219]. When endothelial function was tested, women with endometriosis showed significantly lower FMD compared to controls, although no differences in IMT were found [220]. Thus, endometriosis seems to be related only to a subclinical endothelial dysfunction and atherosclerosis.

### 5.4. Tumor Necrosis Factor-α in Female Fertility

TNF-α has been widely studied in relation to female fertility, both in animals and in humans. In *TNF-α* knockout mouse models, increased proliferation of granulosa cells and decreased oocytes apoptosis were described compared to wild type, confirming that the lack of TNF-α activity increases fertility [221]. In humans, TNF-α has been associated with inflammatory mechanisms related to implantation, placentation, and pregnancy outcome. Although the evidence is still limited, the reduction of TNF-α could be the target of specific therapies in women with obstetric complications, such as recurrent pregnancy loss, early and severe pre-eclampsia, and recurrent implantation failure syndrome [222].

## 6. Endothelial Function and Assisted Reproductive Technology

In light of the increasing evidence describing the connection between endothelial health and reproductive function, several endothelial markers have been evaluated as potential predictors of assisted reproductive technology (ART) outcome. For instance, among these, VEGF-A was identified as a negative predictor of in vitro fertilization (IVF) success, measured in both female blood [223] and embryo culture medium [224]. Moreover, several cytokines have been investigated in an ART setting. TNF-α detected in follicular fluid on the day of oocyte retrieval has been associated with reduced number of fertilized oocytes while serum TNF-α levels were negatively correlated with fertilization rate [225]. Other pro-inflammation cytokines, such as IFN-γ, IL-12, IL-15, IL-18, and MMPs, were demonstrated to be detrimental to reproductive success in women with repeated IVF–embryo transfer failure [226,227,228]. Accordingly, the ART outcome is potentially influenced by cytokine levels detected in the seminal plasma of the male counterpart, suggesting that ART negative outcomes are related to the increased inflammatory status in at least one of the two partners. Contrasting results were provided by studies not detecting different follicular fluid and blood serum concentrations of TNF-α, IL-1β, IL-6, IL-8, IL-12, and NO between women becoming pregnant after ART and those who do not [229,230]. Therefore, agreed opinions on the link between pro-inflammatory cytokines and ART outcomes is not achieved so far. Hitherto, considering the overall evidence, neither serum nor intrafollicular concentrations of these molecules can be considered predictive of the ART outcome.

The involvement of *VEGF* gene SNPs in reproductive functions was previously evaluated [193]. Several *VEGF* SNPs were associated with recurrent spontaneous miscarriage, preeclampsia, and premature birth [231,232]. In addition, the frequency of C allele genotype in *VEGF-A* c.936C > T (rs7664413) and in c.405G > C (rs2010963) were higher in non-pregnant women after IVF than in those pregnant [224,233]. Similarly, the *TNF* c.-308A > G (rs1800629) A allele, associated to a reduced transcriptional activity [233], was linked to higher implantation and pregnancy rates in women undergoing ART [233,234]. Recently, other genes were investigated, trying to obtain an a priori prediction of ART outcomes [235]. As observed in others multifactorial conditions, a combination of SNPs belonging to different genes likely contributes in determining the success of the clinical treatment [233]. In the ART context, we are currently far from the identification of reliable markers.

## 7. Conclusions

Endothelial dysfunction is largely investigated both in clinic and research settings to predict cardiovascular risk. Several molecular markers of endothelial dysfunction are known nowadays and here we described their roles in human fertility [236,237]. However, a real future use of these molecules as reliable markers in the clinical work-up of infertile couples is still far to be achieved. Indeed, previous research attempting to find a correlation between fertility and infertility parameters and endothelial dysfunction detected possible candidates unsuitable or weakly exploitable in clinical practice. The polygenic nature of regulatory mechanisms underlying reproduction prevents to find unique, clear-cut markers of fertility and infertility, which is rather the result of several contributing factors. In conclusion, although a link between endothelial health and reproductive parameters was established, no clear and suitable markers of fertility and infertility are available so far.

## Figures and Tables

**Figure 1 ijms-22-02584-f001:**
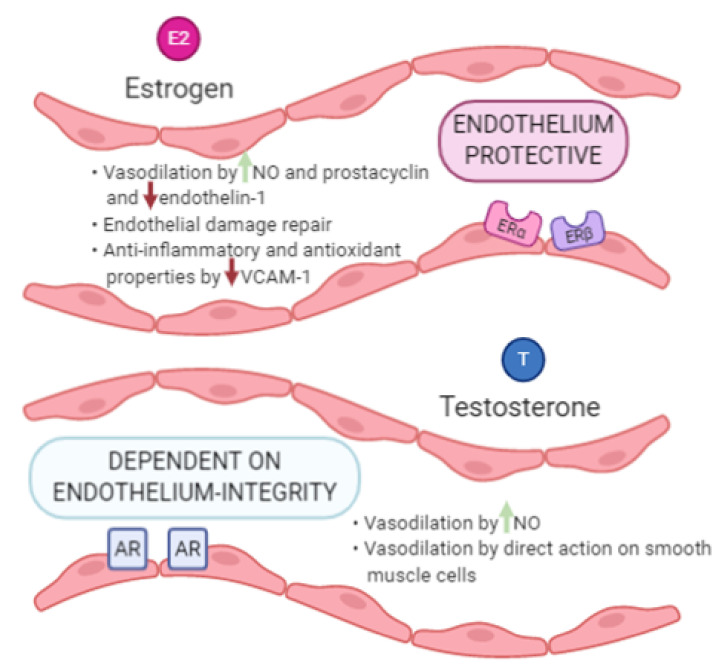
A graphical representation of the synthesis of the estrogens and testosterone actions at endothelial level. Footnote to Figure 1: AR: androgen receptor; ER: estrogen receptor; NO: nitric oxide; VCAM: vascular cell adhesion molecules.

**Figure 2 ijms-22-02584-f002:**
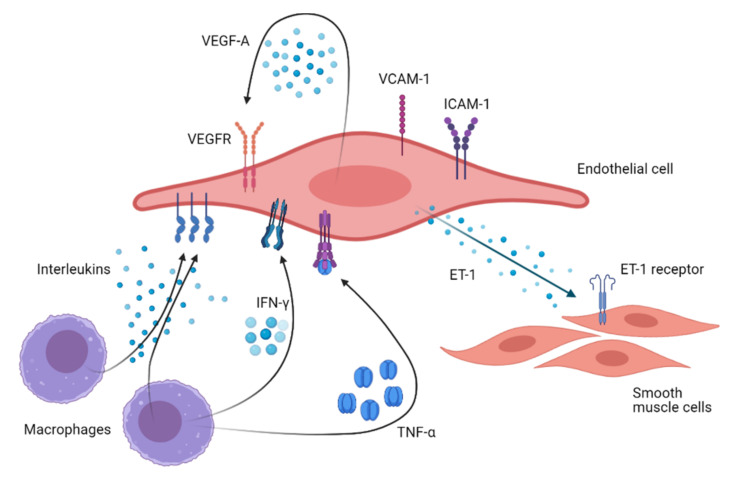
A graphical representation of the paths of endothelial markers involved in male infertility. Footnote to Figure 2: ET-1: endothelin-1; ICAM: intracellular cell adhesion molecules; IFN: interferon; TNF-α: tumor necrosis factor-alpha; VCAM: vascular cell adhesion molecules; VEGF-a: vascular endothelial growth factor-A.

**Table 1 ijms-22-02584-t001:** Markers of endothelial dysfunction applied to clinical practice.

Marker	Target	Test Type	Clinical Application
Ach intracoronary infusions	eNO production: vasodilation	Invasive dynamic	Limited
Non-invasive Ach test	eNO production: vasodilation	Non-invasive dynamic	Limited
FMD	Vasodilation	Non-invasive dynamic	Yes
Carotid IMT	Arterial stiffness	Non-invasive dynamic	Yes
Aortic IMT	Arterial stiffness	Non-invasive dynamic	Yes
Brachial IMT	Arterial stiffness	Non-invasive dynamic	Yes
Common arteries IMT	Arterial stiffness	Non-invasive dynamic	Yes
Pulse wave velocity	Arterial stiffness	Non-invasive dynamic	Limited
Retinal vessels vasodilation assessment	eNO production: vasodilation	Non-invasive dynamic	No
BOLD MRI	Capillary oxygen content	Non-invasive dynamic	Limited
ASL MRI	Tissue perfusion	Non-invasive dynamic	Limited
PC MRI	Blood flow velocity and arterial stiffness	Non-invasive dynamic	Limited
FMD MRI	Vasodilation	Non-invasive dynamic	Limited
Oximetry MRI	Hemoglobin oxygen saturation	Non-invasive dynamic	Limited
Renal resistive index	Arterial elasticity	Non-invasive dynamic	Limited
Dynamic renal resistive index	Arterial elasticity	Non-invasive dynamic	No
LDL	Atherosclerotic plaque development	Biochemical marker	Yes
Oxidative LDL	Atherosclerotic plaque development	Biochemical marker	Yes
Triglycerides	Atherosclerotic plaque development	Biochemical marker	Yes
HDL	Atherosclerotic plaque development	Biochemical marker	Yes
Hs-CRP	Inflammation	Biochemical marker	Yes
Homocysteine	Inflammation	Biochemical marker	Yes
Lipoprotein-associated phospholipase A2	Inflammation	Biochemical marker	Limited

Ach: acetylcholine; ASL: arterial spin labelling; BOLD: blood oxygen level-dependent; eNO: endothelial nitric oxide; FMD: flow-mediated dilation; HDL: high-density lipoprotein; hs-CRP: high sensitivity C-reactive protein; IMT: intima-media thickness; LDL: low-density lipoprotein; MRI: magnetic resonance imaging; PC: phase contrast.

**Table 2 ijms-22-02584-t002:** Markers of endothelial dysfunction commonly used in research setting.

	Soluble Markers	Tissue Markers
**Markers of** **Inflammation**	ADAMs (disintegrin and metalloproteinase) ADAMs with thrombospondin motifs (ADAMTS) -13 Asymmetric dimethyl arginine (ADMA) Bone morphogenetic protein 4 (BMP4) Endothelial progenitor cells (EPCs) Glutathione Interleukin (IL)-1β, -6, -7, -8, -12, -15, -18 Interferon-γ (IFNγ) Long non-coding RNA (LncRNAs) MicroRNAs (miRNAs) N-terminal fragment of proatrial natriuretic peptide (NTpro-ANP) Plasminogen activator inhibitor-1 (PAI-1) Reactive oxygen species (ROS) Sphingosine 1-phosphate Soluble ICAM-1 and VCAM-1 Soluble P-selectin Soluble E-selectin Tumor necrosis factor-α (TNF-α) Von Willebrand factor (VWF)	Angiotensin converting enzyme (ACE) E-selectin Intercellular adhesion molecule-1 (ICAM-1) Metalloproteinases (MMPs) P-selectin Platelet endothelial cell adhesion molecule (PECAM-1) Tissue plasminogen activator (t-PA) Vascular cell adhesion molecule-1 (VCAM-1) Vascular Endothelium (VE)-cadherin
**Markers of** **Vasoconstrictions**	ACE-2 Angiopoetin-1, Angiopoetin-2 Endothelin-1 Sphingosine 1-phosphate Serine carboxypeptidase 1 Thromboxane A2 and its receptor Thrombin, Thrombomodulin Vascular endothelial growth factor-A Von Willebrand factor	ADAMs ADAMTS-1, -4, -7, -13, -18 Advanced glycation end-products (AGEs) E-selectin, P-selectin Fatty acid binding proteins ICAM-1, -2 Neuropeptide Y receptor Thromboplastin (coagulation factor III) Tumor endothelial marker 8 Tissue inhibitor of metalloproteinases (TIMPs) Tissue nonspecific alkaline phosphatase Tumor necrosis factor receptor 2 VCAM-1 VEGF receptor-1
**Markers of** **Vasodilation**	Acetylcholine (Ach) Atrial natriuretic factor (ANF) Bradykinin C-terminal pro-arginine vasopressin C-terminal endothelin-1 precursor fragment Endothelial nitric oxide synthase Nitric oxide (NO) Prostaglandins Serotonin Soluble VE-cadherin Transforming growth factor beta (TGF-β) Tumor necrosis factor α (TNFα)	α1-subunit of the soluble guanylyl cyclase Adenosine Aminopeptidase N Angiotensin receptor (AGTR) 1 Endothelin receptor type B Endothelial cell-selective adhesion molecule IL-1 receptor, IL-13 receptor α1 Integrin β1 Notch Platelet endothelial cell adhesion molecule Prostacyclin receptor VEGF, VEGF receptor 2 VE-cadherin Vasoactive intestinal peptide (VIP) receptor 1,2

**Table 3 ijms-22-02584-t003:** Potential markers of male infertility. The level of evidence is determined through studies available in the literature.

Marker	Category	Effect on Male Fertility	Level of Evidence
c.T786C (rs2070744)	SNP on promoter region of *eNOS*	Increased NO production, inducing direct oxidative damage	Strong
c.G894T (rs1799983)	SNP on exon 7 of *eNOS*	Increased NO production, inducing direct oxidative damage	Mild
Variable number of tandem 4a4b repeats (rs61722009)	CNV in intron 4 of *eNOS*	Increased NO production, inducing direct oxidative damage	Mild
p.Val16Ala (rs4880)	SNP on *SOD2*	Increased seminal levels of H2O2, resulting noxious for sperm quality	Mild
c.G362A (rs2536512)	SNP on *SOD3*	Reduce SOD activity	No evidence
c.C262T (rs1001179)	SNP on *CAT*	Doubled risk of male infertility	No evidence
p.Pro200Leu (rs1050450)	SNP on *GXP1*	Doubled risk of male infertility	No evidence
IL-8	Pro-inflammatory cytokine	Directly related to leucocytospermia and inversely to ejaculatory volume only in patients with male accessory gland infections	No evidence
IL-7	Pro-inflammatory cytokine	Reduced semen quality	Mild
IL-6	Pro-inflammatory cytokine	Increase of pro-inflammatory mediators in the seminal plasma	Contradictory results
IL-1	Pro-inflammatory cytokine	Increase of pro-inflammatory mediators in the seminal plasma	Contradictory results
IFN-γ	Macrophage-activating cytokine	Reduced semen quality	Mild
TNF-α	Pro-inflammatory cytokine	Increase of pro-inflammatory mediators in the seminal plasma	Contradictory results
ICAM-1 and ICAM-2	Adhesion molecule	The balance between the ICAM-1/soluble-ICAM-1 opposite actions influences the permeability of the blood–testis barrier	Mild
VCAM-1	Adhesion molecule	VCAM-1 facilitates the leukocyte adhesion to the vascular endothelium	Mild
ET-1	Vasoconstriction molecule	ET-1 dysfunction could impair sperm progression through the genital tract	Evidence only in animal models
VEGF-A	Gene with pro- and anti-angiogenic properties	VEGF-A isoforms determines a sperm number reduction	Evidence only in animal models

CAT: catalase; CNV: copy number variants; eNOS: endothelial nitric oxide synthase; ET-1: endothelin-1; GXP1: glutathione peroxidase 1; ICAM: intracellular cell adhesion molecules; IFN: interferon; IL: interleukin; SOD: superoxide dismutase; SNP: single-nucleotide polymorphism; TNF-α: tumor necrosis factor-alpha; VCAM: vascular cell adhesion molecules; VEGF-a: vascular endothelial growth factor-A.

## Data Availability

Not applicable.

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
