# Peer review of "The “Hitchhiker’s Guide to the Galaxy” of Endothelial Dysfunction Markers in Human Fertility"

_ijms, 2021, doi:10.3390/ijms22052584_

Round 1
Reviewer 1 Report
The topic of the review article is very relevant.
The review is well written and well structured.
I have no big comments.
Small remarks:
- In the list of references, source No. 114 is not indicated.
- It is necessary to increase the number of sources for 2016-2020, especially for 2018-2020, so that there are at least 30% of the total list.
- It is necessary to make a small correction of the English language in the text.
Author Response
Response to Reviewer 1 Comments
The topic of the review article is very relevant.
The review is well written and well structured.
I have no big comments.
Small remarks:
Point 1: In the list of references, source No. 114 is not indicated.
Response 1: Thank you for your indication. We updated the bibliography, adding missing reference.
Point 2: It is necessary to increase the number of sources for 2016-2020, especially for 2018-2020, so that there are at least 30% of the total list.
Response 2: We updated the bibliography, adding newest manuscripts for each topic evaluated.
Point 3: It is necessary to make a small correction of the English language in the text.
Response 3: The English language was revised throughout the manuscript.
Reviewer 2 Report
Dear Authors,
I found the article by Santi Daniele et al. very interesting. The connection of endothelial dysfunction to human fertility, while intuitively obvious, is not an issue that is often raised in review articles.
Hence, just taking up such a topic is an exciting challenge. The authors coped with it very well. The article is well planned and good written. To the best of my knowledge, it contains all the aspects that should be included in this article and will allow the reader to understand all of the critical elements of the link between endothelium and human fertility.
Of course, each of the subsections could be further elaborated. On the other hand, it could overwhelm the article. Hence, we have two options here (1) accept it as enough or (2) ask that researchers expand on each chapter. But that would greatly expand this review article. I am in favour of not developing these subsections. Because, in fact, almost every subsection could be the subject of an entirely separate article.
I noticed some minor shortcomings that I think could be corrected, e.g. missing in Table 2, IL-18, or metalloproteinases (MMPs); however, Authors wrote in their article:
Other pro-inflammation cytokines as IFN-γ, IL-12, IL-15, IL-18, and MMPs were demonstrated detrimental for reproductive success in women with repeated IVF-embryo transfer failure [219-221].
Maybe it's better to use in the table description - the most commonly used markers… Then we don't have to worry that some marker has been missed?
In addition, I would change the name of Figure 1. There is: A graphical synthesis - should be replaced by A graphical representation of the synthesis ...
Besides, I have no more comments; I believe that after minor additions, the article by Santi Daniele et al. is suitable for publication in such a good and reputable journal as the International Journal of Molecular Sciences.
Best regards
Author Response
Response to Reviewer 2 Comments
Dear Authors,
I found the article by Santi Daniele et al. very interesting. The connection of endothelial dysfunction to human fertility, while intuitively obvious, is not an issue that is often raised in review articles.
Point 1: Hence, just taking up such a topic is an exciting challenge. The authors coped with it very well. The article is well planned and good written. To the best of my knowledge, it contains all the aspects that should be included in this article and will allow the reader to understand all of the critical elements of the link between endothelium and human fertility.
Of course, each of the subsections could be further elaborated. On the other hand, it could overwhelm the article. Hence, we have two options here (1) accept it as enough or (2) ask that researchers expand on each chapter. But that would greatly expand this review article. I am in favour of not developing these subsections. Because, in fact, almost every subsection could be the subject of an entirely separate article.
Response 1: Thank you for your comment. We agree with you that each subsection may deserve a separate article, therefore further development of the topics could make reading the review extremely complicated
Point 2: I noticed some minor shortcomings that I think could be corrected, e.g. missing in Table 2, IL-18, or metalloproteinases (MMPs); however, Authors wrote in their article:
Other pro-inflammation cytokines as IFN-γ, IL-12, IL-15, IL-18, and MMPs were demonstrated detrimental for reproductive success in women with repeated IVF-embryo transfer failure [219-221].
Maybe it's better to use in the table description - the most commonly used markers… Then we don't have to worry that some marker has been missed?
Response 2: the specific aim of the table 2 was to summarize all possible markers used to evaluate endothelial dysfunction. However, we agree with your suggestion and we simplified the table, leaving only most commonly used markers.
Point 3: In addition, I would change the name of Figure 1. There is: A graphical synthesis - should be replaced by A graphical representation of the synthesis ...
Response 3: We agree and changed accordingly the figure 1 caption
Point 4: Besides, I have no more comments; I believe that after minor additions, the article by Santi Daniele et al. is suitable for publication in such a good and reputable journal as the International Journal of Molecular Sciences.
Response 4: Thank you for your suggestions and comments. All changes are reported in the text and written in red as required.